# The *ldp1* Mutation Affects the Expression of Auxin-Related Genes and Enhances SAM Size in Rice

**DOI:** 10.3390/plants13060759

**Published:** 2024-03-07

**Authors:** Zhanglun Sun, Tianrun Mei, Xuan Tan, Tingting Feng, Ruining Li, Sumei Duan, Heming Zhao, Yafeng Ye, Binmei Liu, Aifeng Zhou, Hao Ai, Xianzhong Huang

**Affiliations:** 1Center for Crop Biotechnology, College of Agriculture, Anhui Science and Technology University, Chuzhou 239000, China; sunzl127425@126.com (Z.S.); mtr0322@foxmail.com (T.M.); tx18855072037@126.com (X.T.); fengtingtingyt@ahstu.edu.cn (T.F.); lirn@ahstu.edu.cn (R.L.); duansm@ahstu.edu.cn (S.D.); zhaohm@ahstu.edu.cn (H.Z.); 2Key Laboratory of High Magnetic Field and Ion Beam Physical Biology, Hefei Institutes of Physical Science, Chinese Academy of Sciences, Hefei 230001, China; yyfeng@ipp.ac.cn (Y.Y.); liubm@ipp.ac.cn (B.L.); 3Anhui Xin Fu Xiang Tian Ecological Agriculture Co., Ltd., Maanshan 238200, China; xinfuxiangtian@126.com

**Keywords:** rice, large and dense panicle, RNA-seq, WGCNA, auxin, map-based cloning

## Abstract

Panicle type is one of the important factors affecting rice (*Oryza sativa* L.) yield, and the identification of regulatory genes in panicle development can provide significant insights into the molecular network involved. This study identified a *large and dense panicle 1* (*ldp1*) mutant produced from the Wuyunjing 7 (WYJ7) genotype, which displayed significant relative increases in panicle length, number of primary and secondary branches, number of grains per panicle, grain width, and grain yield per plant. Scanning electron microscopy results showed that the shoot apical meristem (SAM) of *ldp1* was relatively larger at the bract stage (BM), with a significantly increased number of primary (PBM) and secondary branch (SBM) meristematic centers, indicating that the *ldp1* mutation affects early stages in SAM development Comparative RNA-Seq analysis of meristem tissues from WYJ7 and *ldp1* at the BM, PBM, and SBM developmental stages indicated that the number of differentially expressed genes (DEGs) were highest (1407) during the BM stage. Weighted gene coexpression network analysis (WGCNA) revealed that genes in one module (turquoise) are associated with the *ldp1* phenotype and highly expressed during the BM stage, suggesting their roles in the identity transition and branch differentiation stages of rice inflorescences. Hub genes involved in auxin synthesis and transport pathways, such as *OsAUX1*, *OsAUX4*, and *OsSAUR25*, were identified. Moreover, GO and KEGG analysis of the DEGs in the turquoise module and the 1407 DEGs in the BM stage revealed that a majority of genes involved in tryptophan metabolism and auxin signaling pathway were differentially expressed between WYJ and *ldp1*. The genetic analysis indicated that the *ldp1* phenotype is controlled by a recessive monogene (*LDP1*), which was mapped to a region between 16.9 and 18.1 Mb on chromosome seven. This study suggests that the *ldp1* mutation may affect the expression of key genes in auxin synthesis and signal transduction, enhance the size of SAM, and thus affect panicle development. This study provides insights into the molecular regulatory network underlying rice panicle morphogenesis and lays an important foundation for further understanding the function and molecular mechanism of *LDP1* during panicle development.

## 1. Introduction

Rice (*Oryza sativa* L.) is one of the world’s most important food crops, providing a staple food for about half of the global population. The improvement of rice yields is, therefore, crucial to ensure global food safety. The three main factors that constitute rice yield are the number of effective panicles per unit area, the number of grains per panicle, and the 1000-grain weight. However, panicle architecture is a key determinant of the grain number per panicle. Rice panicle architecture varies in panicle length and the number of primary and secondary branches, which are complex quantitative traits mostly controlled by major and minor genes [1,2,3]. Therefore, the exploration of genes related to panicle architecture can provide insights into the regulatory mechanisms underlying variations in panicle types, as well as identify novel gene targets for use in molecular breeding efforts. Following the formation of flag leaves, the shoot apical meristem (SAM) undergoes the transition to the inflorescence meristem with the differentiation of the panicle axis primordia [4]. The development of rice inflorescences primarily consists of five stages, consisting of first bract differentiation, branch meristem initiation, spikelet meristem initiation, organ differentiation, and reproductive cell formation [5].

Recent research into the molecular genetics of rice inflorescence development has been intense and has led to the identification of a network of regulatory genes which governs this process. The activity of meristems and the timing of inflorescence meristem initiation and transition in rice have important impacts on the morphogenesis of rice panicle inflorescences [6,7]. Relative decreases in SAM activity often lead to smaller panicle size and lower yields [5]. Several genes that influence rice meristem activity and rice yield have been identified, including *LAX PANICLE 1* (*LAX1*) [8,9], *LAX2* [10], *CLASS I KNOTTED 1*-*LIKE HOMEOBOX* (*KNOX1*) [11], *ABERRANT PANICLE ORGANIZATION1* (*APO1*) [12,13], *APO2* [14,15,16], *ABERRANT SPIKELET AND PANICLE1* (*ASP1*) [17], and *PANICLE LENGTH 1* (*PAL1*) [18] The loss of functions in these genes usually leads to a decrease in SAM activity in rice, resulting in shorter panicle length, a reduced number of primary and secondary branches, and a decreased number of grains per plant. In addition, *GRAIN NUMBER 1a* (*Gn1a*) negatively regulates the activity of SAM by degrading cytokinins [19]. The dominant mutation in *DENSE AND ERECT PANICLE 1* (*DEP1*) increases meristem activity, resulting in shorter and erect panicles and an increased number of primary and secondary branches, with an increase in grain number [20]. *APO1* and *APO2* synergistically regulate the temporal sequence of rice meristem identity transition [15]. LARGE2 negatively regulates the stability of the APO1 and APO2 complex, and *large2* mutants display an increased SAM size, longer panicles, and an increased number of primary and secondary branches and grain number [3]. FRIZZY PANICLE (FZP) is another negative regulator of APO2, and the loss of function in *fzp* was reported to result in the replacement of spikelets with branches in small panicles. Conversely, the overexpression of *fzp* greatly inhibited the formation of secondary branches, resulting in significantly shorter panicles, decreased number of secondary branches, and a reduced number of spikelets [21,22].

Auxins play an important role in the regulation of inflorescence meristem initiation and growth [23,24,25]. The PIN-FORMED (PIN) protein family regulates the intra- to extra-cellular transport of auxin [26,27,28,29], while PIN1c, PIN1d, and OsPINOID (OsPID) proteins interact with PIN1a and PIN1b to positively regulate the number of tillers and grains per rice panicle [29,30]. OsPIN5b participates in the homeostasis, transport, and distribution of auxin [31], and the reduced expression of *OsPIN5b* increases rice panicle length [28]. Additionally, loss of functional mutants in *OsAUX1*, which encodes an auxin influx carrier protein gene, produced shorter rice inflorescences with fewer branches and spikelets [32].

Although progress in the study of the molecular regulation of rice panicle architecture has been remarkable, our current understanding of this complex and polygenic process remains incomplete and requires further investigation. In particular, the use of panicle type- and panicle development-related mutants to identify further genes affecting rice panicle architecture would be useful. In this study, a *large and dense panicle 1* (*ldp1*) mutant was obtained through ion beam mutagenesis of Wuyunjing 7 (WYJ7) and was shown to markedly affect panicle development. Here, our analysis of the *ldp1* phenotypes suggested it is monogenic and could be mapped to a 1.2 Mb region on the seventh chromosome. Our analysis of the effect of the *ldp1* mutation on early transcriptional changes in the inflorescence meristem and their potential relation to changes in panicle architecture is discussed.

## 2. Results

### 2.1. Agronomic Traits Analysis of the Rice ldp1 Mutants

Agronomic traits data were collected from field-grown wild-type WYJ7 and mutant *ldp1* rice plants during the rice maturity stage (Figure 1A). No significant differences were observed between WYJ7 and *ldp1* plant heights or the number of tillers (Figure 1B,C; *p* > 0.05). However, *ldp1* exhibited a significant relative increase in panicle length (Figure 1D, E; *p* = 1.05 × 10^−3^), number of primary and secondary branches (Figure 1F–H; *p* = 4.81 × 10^−8^ and 2.74 × 10^−15^, respectively), number of grains per panicle (Figure 1I; *p* = 1.66 × 10^−14^), and grain yield per plant (Figure 1J,K; *p* = 8.53 × 10^−6^).

We further measured the grain length (Figure 1L), grain width (Figure 1M), and grain thickness (Figure 1N) of WYJ7 and *ldp1*. The results showed that the *ldp1* mutation had no significant effect on grain length, thickness, or 1000-grain weight (Figure 1O,Q,R, respectively) but effected a significant increase in grain width (Figure 1P; *p* = 1.47 × 10^−2^). 

### 2.2. ldp1 Mutation Increases the Size of Meristem at the Apex of Inflorescence

The initiation of the first bract, known as the bract meristem (BM), marks the starting point of reproductive growth (Figure 2A). Scanning electron microscopy observations revealed that *ldp1* had larger SAMs and BMs compared to WYJ7 (Figure 2B). At the primary branch meristem (PBM) stage, the development of bract hairs (BHs) was observed in *ldp1* (Figure 2C), together with a higher number of PBMs (Figure 2D). During the secondary branch meristem (SBM) stage (Figure 2E), it was observed that relative to WYJ7, *ldp1* formed more SBMs, which were mostly covered by denser BHs (Figure 2F). These results indicate that the *ldp1* mutation leads to the formation of a larger SAM with a significantly increased number of PBMs and SBMs.

### 2.3. RNA-Seq Analysis of the Three Initial Stages of Panicle Development from the SAM in WYJ7 and ldp1

#### 2.3.1. Transcriptome Sequencing and Analysis of Differentially Expressed Genes (DEGs)

Transcriptome sequencing was performed using SAM tissues at the BM, PBM, and SBM stages of panicle differentiation in WYJ7 and *ldp1* plants (Appendix A). A total of 789,249,196 clean reads and 118.39 G clean bases were obtained from 18 samples, with an average of 43,847,178 clean reads per sample. The average Q_30_ sequencing quality value for all samples was 91.60% ± 0.44%, and the average GC content was 49.49% ± 1.39%. The average error rate was 0.03% (Appendix A). Over 90.5% of clean reads successfully and uniquely mapped to the rice genome, resulting in the identification of 22,122 expressed genes and indicating the high quality of the RNA-seq data. Additionally, the squared Pearson correlation coefficient (R^2^) between biological replicates of each sample was greater than 0.8 (Appendix A), indicating the reliability of the subsequent RNA-seq analysis.

FPKM values were initially used to analyze the expression levels of genes at the three developmental stages in WYJ7 and *ldp1* (FPKM > 1). 18,963 genes were found to be constitutively expressed in all six sample sets (Appendix A). In WYJ7, 19,947 genes were constitutively expressed at all three stages of inflorescence meristem development, whereas 359, 168, and 362 genes were specifically expressed in the BM, PBM, and SBM stages, respectively (Appendix A). Similarly, the majority of genes (19,290) were constitutively expressed at all three SAM stages in *ldp1*, but 462, 244, and 469 genes were specifically expressed in the BM, PBM, and SBM stages, respectively (Appendix A). An expression heatmap of the 22,122 genes in the six experimental conditions revealed numerous upregulated and downregulated genes at each stage, with a higher number of upregulated genes in *ldp1* compared to WYJ7 (Appendix A). Analysis of the DEGs in the three stages of WYJ7 showed that there were 150 upregulated genes and 330 downregulated genes in W2 vs. W1, 260 upregulated genes and 185 downregulated genes in W3 vs W2, and the highest number of DEGs in W3 vs. W1, with 480 upregulated genes and 647 downregulated genes (Figure 3A). Analysis of the DEGs in the three stages of WYJ7 showed that there were 150 upregulated genes and 330 downregulated genes in W2 vs. W1, 260 upregulated genes and 185 downregulated genes in W3 vs. W2, and the highest number of DEGs in W3 vs. W1, with 480 upregulated genes and 647 downregulated genes (Figure 3A). In *ldp1*, the comparison of DEGs in the three stages of young panicle meristem developmental revealed 170 upregulated genes and 451 downregulated genes in L1 vs. L2, 358 upregulated genes and 656 downregulated genes in L3 vs. L2, and 580 upregulated genes and 1380 downregulated genes in L3 vs. L1 (Figure 3A), respectively. An analysis of the number of DEGs in each comparison in WYJ7 revealed only 24 DEGs common to all three SAM developmental stages (Appendix A), 12 of which were upregulated and 12 were downregulated (Appendix A). A similar analysis of *ldp1* DEGs indicated 48 DEGs common to all three SAM developmental stages (Appendix A), of which 33 were upregulated and 15 downregulated (Appendix A).

To identify DEGs altered in the *ldp1* mutant, the DEGs at each developmental stage in *ldp1* and WYJ7 were compared. This comparison between the three stages of WYJ7 and *ldp1* in the same stage indicated that the number of upregulated DEGs in L1 vs. W1 and L2 vs. W2 was greater than the downregulated genes. The comparison indicated that the *ldp1* mutation had the largest effect on gene expression (1065 genes) in the BM stage (1407 DEGs; Figure 3B), 1065 of which were specific to the BM stage (Figure 3C) and the majority of which were upregulated relative to WYJ7 (Figure 3B). SAMs of *ldp1* mutant plants also showed substantial changes in gene expression in the two following stages of development (Figure 3B), with 472 and 626 specific to the PBM and SBM stages, respectively. An inspection of 83 shared DEGs between WYJ7 and *ldp1* were hierarchically clustered to visualize and identify differences in expression, which revealed a higher number of upregulated genes in *ldp1* compared to WYJ7 (Figure 3D). The data indicate that the initiation of panicle development in SAM tissues occurs with substantial changes in gene expression in WYJ7 but that a considerable portion of these changes are altered in the *ldp1* mutant, which could contribute to the altered morphological development observed.

#### 2.3.2. Analysis of WGCNA and Gene Coexpression Network

Next, we constructed a gene network for all expressed genes across the six experimental conditions using the weighted gene coexpression network analysis (WGCNA) method [33]. Preliminary analyses indicated that with a soft threshold of 0.85 and the weight parameter β set to 7, the network approximated a power law distribution (Appendix A). These parameters were, therefore, used to calculate a topological overlap matrix for every pair of genes. Using a dynamic tree cut algorithm, 31 expression modules were obtained (Figure 4A). A correlation analysis between the genes in different expression modules and the WYJ7 and *ldp1* phenotypes revealed that the expression patterns of genes in the orange and turquoise modules were positively correlated with the *ldp1* phenotype during the three primordial stages of panicle differentiation. In contrast, the expression patterns of genes in the magenta module were negatively correlated with the *ldp1* phenotype during this period (Appendix A). The analysis of the feature vectors of genes (the overall expression pattern of a module or cluster of genes) in these modules indicated a strong positive correlation between the orange and turquoise modules, as well as a strong negative correlation between the orange and turquoise modules and the magenta module (Figure 4B). Analysis of the expression patterns of genes in the orange, turquoise, and magenta modules revealed that the turquoise module was predominantly highly expressed during the BM phase of *ldp1* (Figure 4C), which is consistent with the high number of DEGs observed during this phase (Figure 3C).

A GO enrichment analysis of DEGs within the turquoise module showed an enrichment in genes associated with biological processes related to aspects of ion transport (ion transport, metal ion transport, cation transport), and for molecular functions related to hydrolase activity acting on glycosyl bonds, hydrolase activity hydrolyzing O-glycosyl compounds, transferase activity, and transferring glycosyl groups. No significant enrichment was observed for any cellular component (Appendix A). A KEGG enrichment analysis (Figure 4D) indicated genes associated with the pathways of plant hormone signal transduction were most enriched (osa04075). Tryptophan metabolism (osa00380), N-glycan biosynthesis (osa00510, osa00513), and the TCA cycle (osa0002) were also highly enriched.

We next used five genes related to the auxin signaling pathway and 18 genes related to the tryptophan metabolism pathway as candidate target genes in the turquoise module to draw a gene network diagram. The results suggested that eight of these genes interacted with genes of the turquoise module (Figure 4E). Only one gene of the tryptophan metabolism pathway, aldehyde dehydrogenase (OsALDH3E2), was indicated as a potential interactor of the module gene, *AUXIN INFIUX CARRIER 4* (*OsAUX4*). However, the genes related to the auxin signaling transduction pathway, *OsAUX1*, *SMALL AUXIN UP RNA 25* (*OsSAUR25*), *OsAUX4*, *OsAUX2*, *OsSAUR5*, *INDOLE*-*3*-*ACETIC ACID 27* (*OsIAA27*), and *OsAUX3*, generated 165, 104, 73, 71, 64, 14, and 1 node, respectively (Figure 4E).

#### 2.3.3. GO and KEGG Enrichment Analysis of DEGs during the BM Stage

The above results (Figure 3B) indicated that the *lpd1* induced the largest number of changes in gene expression in the BM stage. A GO functional enrichment analysis of these DEGs indicated enrichment in the biological processes of response to stress, response to oxidative stress, cell wall organization and cell wall modification. There was also gene enrichment for cellular components including the extracellular region and cell wall (extracellular region, cell wall, cell periphery), with lesser enrichments in cytoskeleton-related terms. The most enriched molecular functions were transcription factor and transferase activity (DNA-binding transcription factor activity, transcription regulator activity, transferase activity, transferring glycosyl groups; Appendix A). Furthermore, KEGG pathway analysis of the 89 DEGs common to *ldp1* and WYJ7 at this stage indicated the enrichment of phenylpropanoid biosynthesis (osa00940) as the most enriched pathway (Figure 5 and Appendix A). Other enriched pathways included those which could be related to hormone biosynthesis, including *α*-linolenic acid metabolism (osa00592; jasmonic acid biosynthesis), tryptophan metabolism (osa00380; auxin synthesis), zeatin biosynthesis (osa00908; in cytokinin biosynthesis), cysteine and methionine metabolism (osa00270), and plant hormone signal transduction (osa04075) (Figure 5 and Appendix A). 

Next, we further performed KEGG enrichment analysis (Appendix A) of the 83 DEGs between WYJ7 and *ldp1,* which were common to the three developmental stages of SAM studied (Figure 3C). The results showed that the 83 DEGs were also enriched in pathways including tryptophan metabolism (osa00380), phenylpropanoid biosynthesis (osa00940), and valine, leucine, and isoleucine biosynthesis (osa:4339583). Together, these results suggest that the *ldp1* mutation may affect the differentiation of panicle primordia by affecting endogenous hormones, especially auxin signaling transduction. Figure 6A shows the tryptophan metabolism and auxin signaling pathways related to auxin synthesis and the relative changes in gene activities in these pathways in *ldp1* and WYJ7 at the three SAM development stages (Figure 6B). The RNA-seq data for these genes showed that relative to WYJ7, *OsALDH3E2*, *OsAO2* (a member of the aldehyde oxidase (AO) family), *OsAUX5*, *OsIAA1*, *OsIAA3*, *OsIAA23*, *OsGH3*-*9* (encodes a GH3 family protein), and *OsSAUR22* all showed significant upregulation in the BM stage in *ldp1*, whereas *OsSAUR23* and *OsSAUR27* showed significant downregulation at this stage. The *OsAO2* gene was upregulated in all three stages of panicle differentiation in *ldp1* (Figure 6B). Moreover, qRT-PCR assays confirmed that the relative expression levels of these nine genes during inflorescence SAM development in *ldp1* and WYJ7 were consistent with their expression trends observed from RNA-Seq data (Figure 6C).

### 2.4. Genetic Analysis of the Rice ldp1 Mutant Phenotypes and Gene Mapping of the LDP1 Gene

To genetically characterize the *ldp1* trait, *ldp1* was crossed with the rice variety MH63. The resultant F_1_ plants displayed phenotypes consistent with wildtype MH63. In the F_2_ population, 770 plants exhibited the MH63 panicle phenotype, and 281 individual plants exhibited the *ldp1* phenotype (Appendix A). The chi-square test and the fitness analysis confirmed that the *ldp1* phenotype conforms to a 3:1 segregation ratio (χ^2^ = 1.599 < 3.84), suggesting that the *ldp1* mutant phenotypes were controlled by a recessive monogene.

To roughly map the *ldp1* mutation, 20 individuals of the F_2_ population with a strong *ldp1* phenotype were studied with 68 SSRs, which indicated that *LDP1* was located between the L5 and L7 markers on chromosome 7 (Figure 7A). Subsequently, InDel primers corresponding to the region between these markers were designed (Appendix A) and used to further narrow down the location of the *LDP1* gene between primers L5 and L13 (Figure 7B). An analysis of the rice genome (http://rice.plantbiology.msu.edu/ (accessed on 10 December 2022)) indicated the distance between these markers was 1.23 Mb and contained a total of 166 annotated genes (Appendix A).

### 2.5. Analysis of the Candidate Genes

In order to examine the 166 candidate genes identified in the region containing *LDP1*, these were cross-referenced with DEGs identified from RNA-Seq data. Four DEGs in the fine-mapped region (Figure 7C) were confirmed to be differentially expressed during early panicle development by qRT-PCR (Figure 7D). These were *OsbHLH111*, which encodes a BASIC HELIX-LOOP-HELIX (bHLH) transcription factor, *EPSTEIN*–*BARR VIRUS NUCLEAR ANTIGEN 1*-*BINDING PROTEIN 2* (*OsEBP2*), *OsACTPK2* (a serine/threonine/tyrosine protein kinase containing the ACT domain, also referred to as *ACTPK*), and *OsUG709C2,* which encodes a UDP-glucosyltransferase. However, no mutation sites were found in the exonic sequences of either of these genes.

## 3. Discussion

Since the 1990s, scientists have wanted to breed new or ideal plant architecture (IPA) in rice to break through yield limits [34]. Important features of IPA include fewer non-productive tillers with sturdier stems; leaves which are more erect, thicker and longer; a higher number of grains per panicle (the ideal panicle type, i.e., IPT, for *japonica* rice is dense and erect panicle); and medium plant height for higher harvest index [34,35]. In recent years, the identification of key genes related to important agronomic traits in rice and the dissection of their molecular mechanisms have greatly accelerated the molecular breeding for rice IPA. Of these, the IPT is a crucial factor in determining rice yield potential. However, IPTs can differ between geographical environments. For instance, the acquisition of the high-yield gene *DEP1* typically leads to dense and erect panicle structures in *japonica* rice, and this favorable trait is better suited to an environment [20]. Conversely, the long and drooping inflorescences of *indica* rice are more suitable for fruiting in environments of high temperature and humidity. Therefore, exploring various genes that regulate panicle development according to local conditions can further advance tailored breeding efforts. Due to the limited genetic variation available in modern, highly selected rice cultivars, the modification of promising genes through artificial mutation is an option gaining increasing attention from breeders [36]. The large and dense panicle mutant *ldp1* was obtained from ion beam mutagenesis of the *japonica* rice variety WYJ7. Compared with WYJ7, *ldp1* showed no significant differences in plant type, grain length, grain thickness, and 1000-grain weight (Figure 1). However, this mutant exhibits significant increases in panicle length, the number of primary and secondary branches, the number of grains per panicle, and grain yield per plant (Figure 1). During inflorescence differentiation, *ldp1* produced a larger SAM volume, which is similar to that observed in mutations such as *DEP1* and *LARGE2* [3,20]. The larger overall size of the inflorescence SAM in the *ldp1* mutant suggests that the increased formation of PBM and SBM primordia occurs without any deleterious effects on meristematic center maintenance (Figure 2).

Additionally, during the three initial developmental stages of rice panicle differentiation from the meristem, *ldp1* showed a large number of DEGs compared to WYJ7 (Figure 3). WGCNA and gene coexpression network analysis (Figure 4 and Appendix A) showed that one module of genes (turquoise) was highly and positively correlated with the *ldp1* mutant phenotype and was enriched in genes that were highly expressed during the BM stage. In *Arabidopsis thaliana*, the negative feedback regulation of *WUSCHEL*-*CLAVATA* (*WUS*-*CLV*) mediates the maintenance function of SAM stem cells [37,38,39]. In rice, the orthologous genes of *CLV*, *FLORAL ORGAN NUMBER 1* (*FON1*) [40], *FON2*/*FON4* [41,42], as well as *FON2*-*LIKE CLE PROTEIN1* (*FCP1*) [43], *LONELY GUY* (*LOG*) [44], and *ORYZA SATIVA HOMEOBOX1* (*OSH1*) [45,46] regulate the activity and size of SAM, ultimately affecting panicle architecture, suggesting that the expression of a large number of genes during the SAM period plays a crucial role in rice young panicle differentiation and morphogenesis. A GO and KEGG enrichment analysis of the turquoise module genes suggested that the *ldp1* gene could regulate rice panicle development through its effects on auxin biosynthesis and signal transduction (Figure 4 and Appendix A). The results of the coexpression network analysis of genes related to auxin synthesis and signal transduction indicated that the candidate target genes *OsAUX1* and *OsSAUR2* were the main hub genes, while *OsAUX4*, *OsAUX2*, *OsSAUR5*, *OsIAA27*, *OsAUX3*, and *OsALDH3E2* formed secondary hubs (Figure 4). *OsAUX1* [47], *OsAUX3* [48], and *OsAUX4* [49] have been reported to regulate primary root development and the number of lateral roots in rice. The *OsAUX1* mutants created through CRISPR-Cas9 and T-DNA insertion showed shorter panicles and a reduced number of tillers [32]. *OsAUX3* negatively regulates grain length and weight in rice [50]. *OsSAUR2* and *OsSAUR5*, along with other genes in their families, redundantly regulate the growth and development of rice [51]. The functions of *OsIAA27* and *OsALDH3E2* currently remain unclear. However, *OsIAA27* shows specific expression in rice inflorescence and stems [52].

Enrichment analysis was performed on the *ldp1*-associated DEGs identified in the BM stage, including those common to the three developmental stages of SAM studied. The results suggested possible relationships between altered tryptophan biosynthesis and SAM size. The *ldp1* mutation may cause differential gene expression in the tryptophan metabolism and auxin signal transduction pathways, which could contribute to the increased SAM size observed (Figure 5, Figure 6, Appendix A). In support, the expression levels of *OsALDH3E2* and *OsAO2* were significantly upregulated, which could favor the synthesis of indole-3-pyruvic acid, tryptamine, and indole-3-acetaldehyde oxime, which would favor the generation of indole-3-acetic acid (Figure 6). The synthesized auxin is imported into the nucleus by the auxin transporter protein AUX1 [32], after which it binds to the SCF^TIR1/ARF^ complex, which is an E3 ubiquitin ligase complex promoting the interaction of SCF^TIR1/ARF^ with the complex of auxin response factors AUX/IAA and ARF. This leads to the degradation of the transcriptional repressor protein AUX/IAA within the complex, activating the activity of the ARF transcription factors. The activated ARF transcription factors then bind to auxin response elements (AuxRE), facilitating the transcription of target genes, such as *AUX*/*IAA*, *GH3*, and *SAUR*, which are known to be induced by auxin [53,54]. The upregulated expression of *OsAUX5*, *OsIAA1*, *OsIAA3*, *OsIAA23*, *OsGH3*-*9,* and *OsSAUR22* further suggest an increased activation of the typical SCF^TIR1/ARF^-mediated auxin signaling pathway [53,54].

The genetic analysis indicated that the *ldp1* phenotype is controlled by a recessive monogene (Appendix A) located in a 1.2 Mb locus on the seventh chromosome. A few genes in this region that were also differentially expressed between *ldp1* and WYJ7 at the BM stage were selected for sequencing as candidate genes (Figure 7). However, these failed to show any nucleotide sequence differences between WYJ7 and *ldp1*. Further sequencing analysis of the promoter region of the candidate genes is needed in the future. In addition, it is necessary to use larger isolated populations for further fine mapping. Gene editing techniques could also be used to knock out or overexpress the *LDP1* or hub genes, as well as the identifications of the upstream and downstream interacting proteins of LDP1, its specific functions, and molecular mechanisms.

## 4. Conclusions

In summary, this study demonstrates that the *ldp1* mutant leads to substantial changes in WYJ7 gene expression in the earliest stages of floral meristem development, with consequences for panicle morphology. The *ldp1* mutation increased the size of the SAM and resulted in more primary and secondary branches. Key genes in the auxin signaling pathway were differentially expressed during the three developmental stages of young inflorescence between WYJ7 and *ldp1*. These results imply that *ldp1* does indeed play a regulatory role in the formation of these primordia, but whether this may well be completely independent or dependent of its effect on auxin biosynthesis and signaling requires further investigation. This study lays the foundation for an in-depth understanding of the molecular function/s of the *LDP1* gene and its effects on the regulatory network that determine rice panicle architecture. This work also provides candidate intervals for developing the construction of IPT in rice within molecular breeding approaches.

## 5. Materials and Methods

### 5.1. Planting, Sampling, and Agronomic Trait Investigation of Rice Materials

The *ldp1* rice mutant was obtained as previously described [55]. In May 2021, the seeds of WYJ7 and *ldp1* were germinated under dark conditions at 37 °C. After sprouting, the seeds were spread on a seedling bed at the Anhui Science and Technology University, Fengyang Campus, Anhui Province (32°52′30″ N, 117°33′15″ E). Once the seedlings reached the growth stage of four leaves and one heart, they were transplanted into rice fields consisting of equal fertility and similar soil textures. The agronomic traits of WYJ7 and *ldp1* were observed throughout the growth period. The agronomic traits investigated included number of tillers, primary and secondary branches, and grains per panicle; panicle length; plant height; grain yield per plant; grain length; grain width; grain thickness; and 1000-grain weight.

The transition of the SAM from vegetative to reproductive growth is marked by the differentiation of the first bract primordium (stage I), followed by formation of the primary (stage II) and secondary branch primordia (stage III). During this period, SAMs were dissected from the shoot apex under a dissecting microscope. Samples for scanning electron microscopy were fixed in a solution of glycol and stored at 4 °C. For RNA-seq analysis, three biological replicates were prepared during each stage, resulting in a total of 18 samples, which were flash-frozen in liquid nitrogen and stored at −80 °C until analysis. In this study, the SAM samples from WYJ7 and *ldp1* were named W1–3 and L1–3, respectively.

At the mature stage of rice, the agronomic traits of WYJ7 and *ldp1* rice were measured. Statistical analysis of the data was performed using Student’s *t*-test in SPSS software (IBM SPSS Version 21.0), and graphs were generated using GraphPad Prism software (FreeImage Public License-Version 1.0).

### 5.2. Scanning Electron Microscopy

The tissues were fixed in a solution of butanediol (Wuhan Servicebio Co., Ltd., Wuhan, China), then dehydrated in a series of ethanol gradients (30%, 50%, 70%, 80%, 90%, 100%) for 15–30 min each. The samples were then placed in 100% isoamyl acetate for 30 min and dried at the critical point of CO_2_. Subsequently, the samples were mounted and coated with gold. The samples were observed using a Hitachi S-2460 scanning electron microscope (Hitachi, Tokyo, Japan).

### 5.3. RNA-Seq and WGCNA Analyses

Total RNA was extracted using the TIANGEN RNAprep Pure Plant Plus Kit (TIANGEN, Beijing, China) following the provided protocol. The RNA integrity was assessed using the Agilent 2100 bioanalyzer (Agilent, Santa Clara, CA, USA). The cDNA library construction and sequencing were entrusted to Beijing Novogene Co., Ltd. (Beijing, China). The clustering of the index-coded samples was performed on a cBot cluster generation system using TruSeq PE Cluster Kit v3-cBot-HS (Illumina, Beijing, China) according to the manufacturer’s instructions. After cluster generation, the library preparations were sequenced on an Illumina Novaseq platform, and 150 bp paired-end reads were generated. The data were quality controlled using fastp software (version 0.19.7) with parameters fastp-g-q5-u50-n15-l150, and the raw reads with sequencing adapters or low-quality sequences were removed. The quality-controlled clean reads were aligned to the reference genome *japonica* rice variety Nipponbare IRGSP-1.0 version (https://ftp.ncbi.nlm.nih.gov/genomes/all/GCF/001/433/935/GCF_001433935.1_IRGSP-1.0 (accessed on 10 December 2022)) using HISAT2 software (v2.0.5) [56].

The gene expression levels in RNA-seq were represented by FPKM (fragments per kilobase of transcript per million mapped reads) values. Differentially expressed genes (DEGs) were selected based on the thresholds of |log_2_^fold change^| ≥ 1 and *p* adjust ≤ 0.05. Venn diagrams were generated to visualize the coexpression of genes between the 6 sample sets. GO (Gene Ontology) and KEGG (Kyoto Encyclopedia of Genes and Genomes) enrichment analyses were performed on the DEGs as previously described [57]. GO enrichment included biological process (BP), cellular component (CC), and molecular function (MF) [58], with the top 10 significant terms selected for each category. KEGG enrichment integrated genomic, chemical, and systemic functional information [59], with the top 20 significantly enriched pathways selected for analysis.

The WGCNA was conducted using the R package in the Novogene Cloud Platform, following the methods previously described [33]. Genes with a Pearson correlation coefficient greater than 0.8 were used, with an unsigned topological overlap matrix, a merge cut height of 0.75, and a minimum gene number of 30 in each module. Genes with similar expression patterns were grouped into different modules. The association between modules and traits was estimated by correlating their gene expression patterns with meristem tissue type and/or the WYJ7 and *ldp1* background. Cytoscape [60] software (v3.10.1) was used for visualization analysis of the gene network, with a node weight > 0.15. Hub genes in each module, which were genes that produced a high number of nodes in the module, i.e., those containing genes most affected by the *ldp1* mutation, were selected.

### 5.4. Quantitative Real-Time PCR (qRT-PCR) Analysis

MonScript™ RTIII All-in-One Mix with dsDNase (Suzhou Monna Biotech, Suzhou, China) was used to prepare cDNA following the instructions provided. qRT-PCR assays were prepared using MonScript™ ChemoHS Specificity Plus qPCR Mix (Low ROX) (Suzhou Monna Biotech) according to the manufacturer’s instructions with the gene-specific primers given in Appendix A. The qRT-PCR amplification was performed on an ABI ViiA7 Real-Time PCR System (Life Technologies, Carlsbad, CA, USA) in a 10 µL reaction volume containing 2 µL cDNA template (20 ng/μL), 5 μL MonAmpTM chemoHS qPCR Mix (Suzhou Monna Biotech), 0.1 μL Low ROX Dye (100×), and 10 µmol/L of each primer. The reaction procedure was as follows: denaturation at 95 °C for 10 min, followed by 40 cycles of denaturation at 95 °C for 10 s and annealing/extension at 60 °C for 30 s. The expression levels of selected genes were calculated relative to *Actin*^WYJ7^ using the 2^−ΔΔCT^ method [61] with three biological replicates.

### 5.5. Construction of F_2_ Segregating Population and Mapping of the LDP1 Gene

The mutant *ldp1* was crossed with the conventional *indica* rice variety Minghui 63 (MH63) to obtain F_1_ generation seeds. The F_1_ plants were self-crossed to obtain the F_2_ generation segregating population. At maturity, the segregation ratio of MH63 and *ldp1* phenotypes was determined using a chi-square test calculated with SPSS (IBM SPSS Version 21.0) software.

Individuals showing the *ldp1* phenotype in the F_2_ segregating population and F_2:3_ families were selected as candidates for gene mapping. The DNA of young leaves was extracted using the improved cetyltrimethylammonium bromide (CTAB) method [62]. Initially, 64 pairs of simple sequence repeats (SSRs) distributed across the 12 chromosomes of rice were selected. PCR amplification and 4% agarose gel electrophoresis analysis were conducted between the parents, resulting in 58 polymorphic primers. Subsequently, 20 typical individuals showing *ldp1* phenotypes were used for preliminary gene mapping. InDel primers were designed (Appendix A), synthesized by Anhui General Biological Co., Ltd., and used for fine mapping of the *LDP1* gene using 261 individuals showing *ldp1* phenotypes. The PCR amplification system was conducted in 20 μL, containing 2 μL (25 ng/μL) of DNA template, 10 μL of 2 × Santaq PCR mix (Nanjing Novozymes Biological, Nanjing, China), and 0.5 μL (10 µmol/L) of each primer. The amplification process included an initial denaturation at 95 °C for 3 min; 35 cycles of denaturation at 95 °C for 15 s, annealing at 55 °C for 15 s, extension at 72 °C for 5 s; and a final extension at 72 °C for 10 min. The PCR products were electrophoresed on 4% agarose gels and visualized under UV light.

### 5.6. Candidate Gene Analysis

Genes within the mapping interval were analyzed, then those genes differentially expressed on RNA-seq data in WYJ7 and *ldp1* were selected for sequencing and qRT-PCR validation. Gene-specific primers were designed (Appendix A) for PCR amplification. The PCR products were sequenced (Anhui General Biological Co., Ltd., Hefei, China) and aligned with the reference genome sequence of the Nipponbare 7.0 version using SnapGene (GSL Biotech LLC 6.0.2.0) software to select the candidate *LDP1* gene.

## Figures and Tables

**Figure 1 plants-13-00759-f001:**
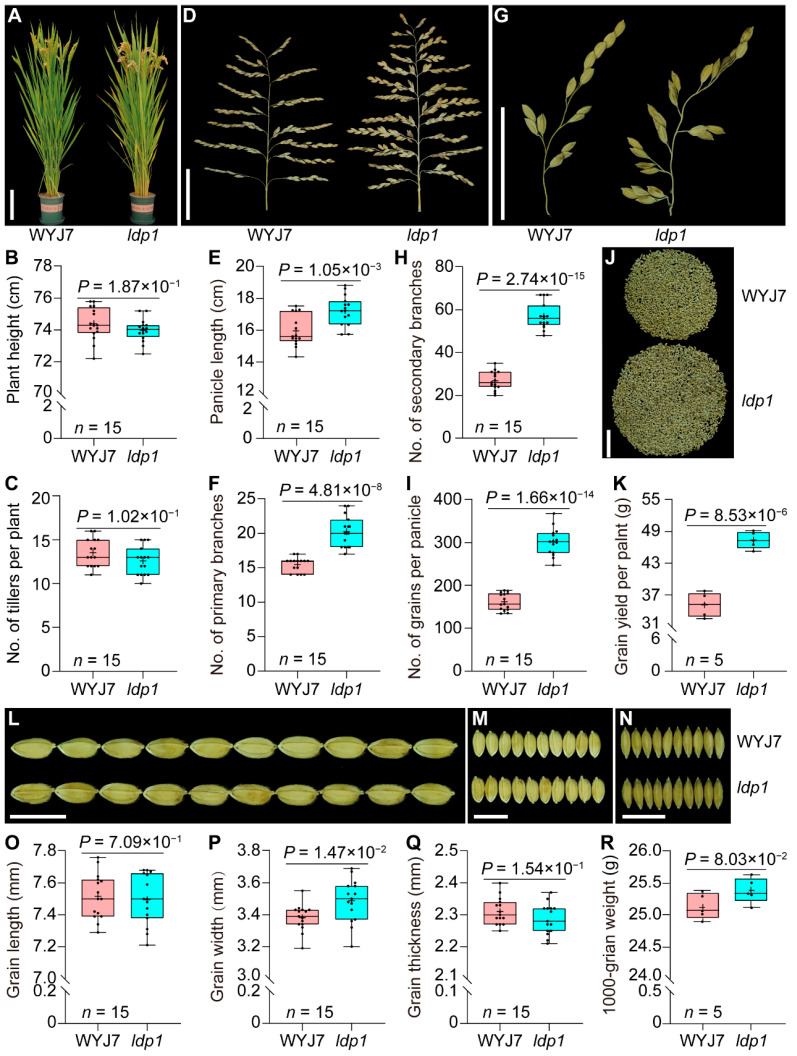
Comparison of WYJ7 and the *ldp1* agronomic traits. (**A**,**B**) Plant height; (**C**) tiller number; (**D**) panicle type; (**E**) panicle length; (**F**) number of primary branches; (**G**,**H**) number of secondary branches; (**I**) number of grains per panicle; (**J**,**K**) grain yield per plant; (**L**,**O**) grain length; (**M**,**P**) grain width; (**N**,**Q**) grain thickness; (**R**) 1000-grain weight. Scale bars in panels (**A**), (**D**,**G**,**J**), and (**L**,**M**,**N**) are 20 cm, 5 cm, and 1 cm, respectively.

**Figure 2 plants-13-00759-f002:**
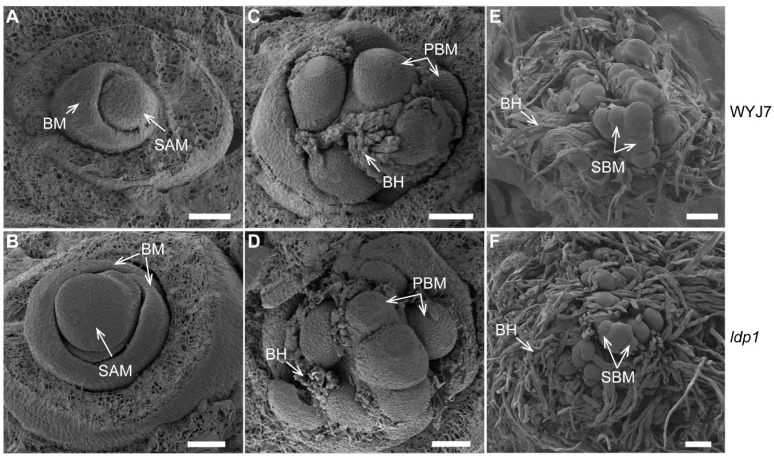
SEMs of rice inflorescence meristems at three early stages of panicle formation in WYJ7 and *ldp1*. (**A**,**B**) The transition of the SAM (shoot apical meristem) to inflorescence growth with BM (bract meristem) formation (BM stage); (**C**,**D**) primary branch meristem (PBM) and bract hair (BH) formation (PBM stage); (**E**,**F**) secondary branch meristem (SBM) and BH formation (SBM stage). Scale bar: 50 μm.

**Figure 3 plants-13-00759-f003:**
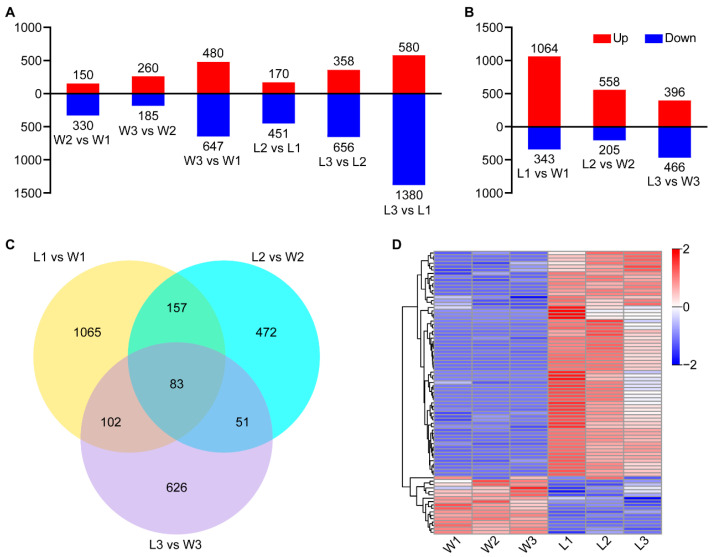
A comparison of the DEGs occurring at different stages of panicle development in *ldp1* and WYJ7 rice. (**A**) The number of the up- and downregulated DEGs at the three stages of young panicle development in WYJ7 and *ldp1*; (**B**) the impact of the *ldp1* mutation on DEGs in WYJ7 at the three initial stages of panicle development; (**C**) the distribution of DEGs affected by *ldp1* across the three initial stages of panicle development, showing varying degrees of specificity; (**D**) expression features cluster analysis of the shared 83 common DEGs across three stages of young panicle development in WYJ7 and *ldp1*. The clustering method adopts hierarchical clustering, horizontally clustering genes with similar expression patterns (FPKM values close) horizontally, and normalizes the rows using Z-scores; the stages W1–3 and L1–3 indicate the three developmental stages of BM, PBM, and SBM in WYJ7 and *ldp1*, respectively. Significance threshold for DEGs: |log_2_^fold change^| ≥ 1, *p* < 0.05.

**Figure 4 plants-13-00759-f004:**
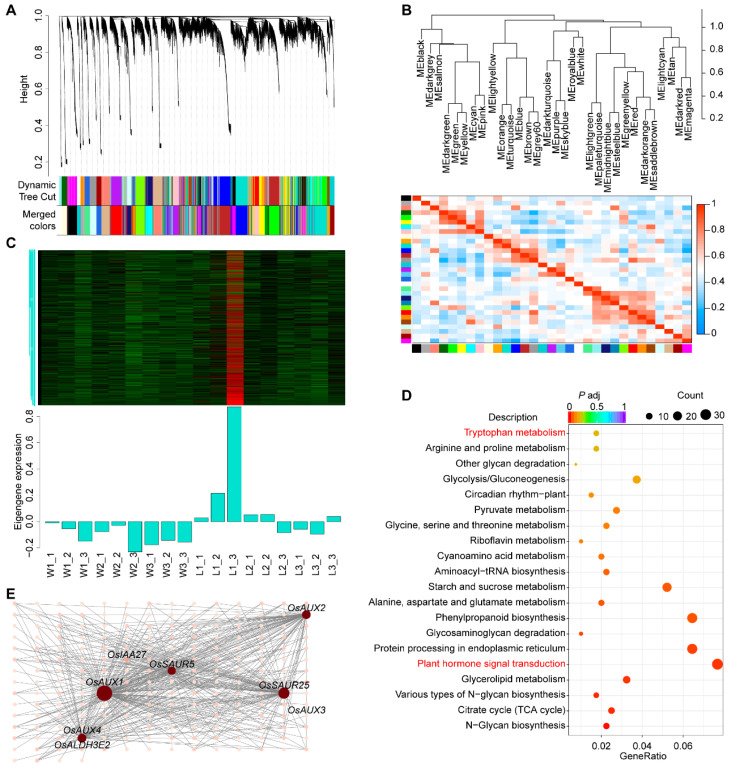
WGCN analysis of RNA-seq data during the BM stage of panicle formation in WYJ7 and *ldp1*. (**A**) Identification of the 31 modules using correlations of gene coexpressions, hierarchical clustering, and dynamic tree cut; (**B**) hierarchical cluster tree showing the dissimilarities between the 31 gene expression modules (**upper** panel) and heat map of inter-module correlations (**lower** panel); (**C**) gene expression pattern of the turquoise module; (**D**) KEGG enrichment analysis of genes in the turquoise module, with red labels indicating the pathways of plant hormone signal transduction and tryptophan metabolism; (**E**) coexpression network diagram of genes involved in auxin synthesis and signal transduction. The degree of connectivity was set at >0.15. W1–3 and L1–3 are as described in Figure 3. The suffix _1–3 in these labels represents the three independent biological replicates.

**Figure 5 plants-13-00759-f005:**
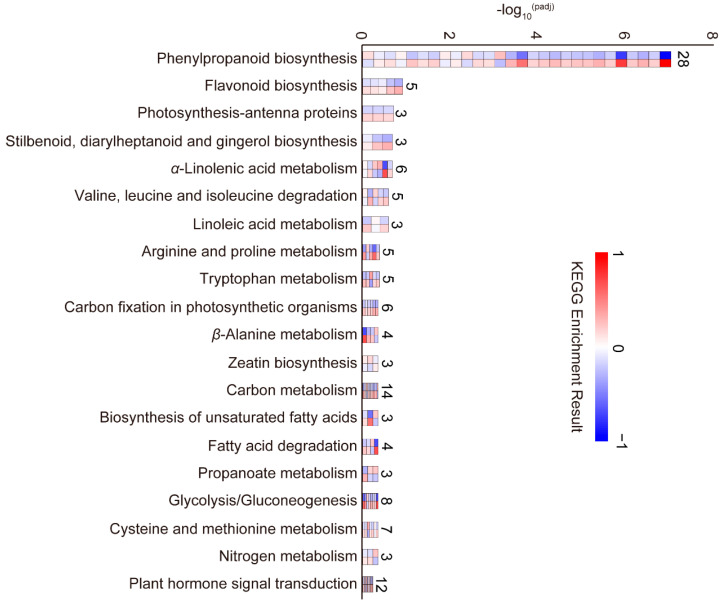
KEGG pathway alterations in response to the *ldp1* mutation in the BM stage of panicle primordia formation. The pathways affected are indicated to the left. The number of genes involved in each pathway is indicated to the right. The histogram bars are colored to represent the expression heat maps of the pathway-associated DEGs in the SAM samples, where W1 is the top row and L1 is the lower row. The color scale used is given on the right.

**Figure 6 plants-13-00759-f006:**
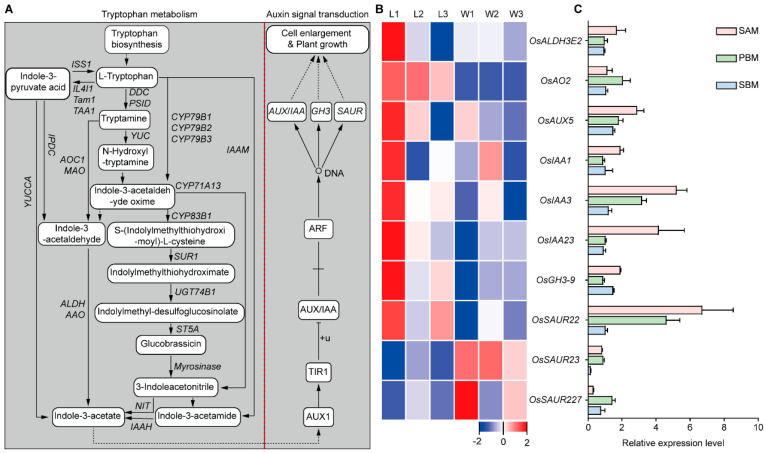
Differential expressions of the DEGs involved in the tryptophan metabolism and auxin signal transduction. (**A**) Schematic diagram of the indole-3-acetic acid synthesis and auxin signal transduction; (**B**) heat map depicting the expression profile of the nine DEGs at the three developmental stages in *ldp1* and WYJ7; (**C**) qRT-PCR analysis of the relative expression levels (2^−ΔΔCT^) of the DEGs in *ldp1* compared to WYJ7. W1–3 and L1–3 are as in Figure 3. BM, PBM, and SBM indicate the three developmental stages of the SAM in *ldp1* and WYJ7.

**Figure 7 plants-13-00759-f007:**
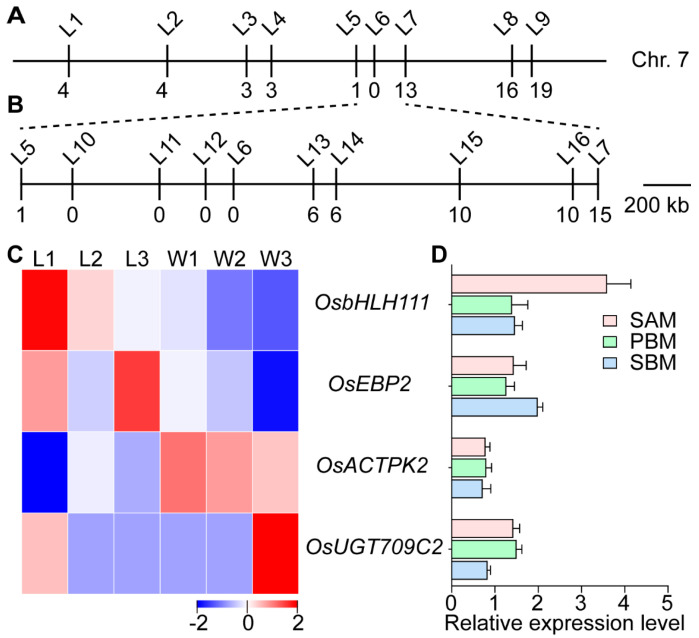
Fine mapping and candidate gene analysis of the *ldp1* mutation. (**A**) Preliminary mapping of *LDP1*. The number below the SRR/InDel marker (L1–L16) represents the number of plants identified with that marker; (**B**) Fine mapping of *LDP1* using designed InDels and 261 individual plants; (**C**) heat map depicting the expression profile of the five DEGs in six tissues; (**D**) qRT-PCR analysis of the relative expression levels of four genes in the fine-mapping interval between WYJ7 and *ldp1* at the three developmental stages of young panicle development. W1–3 and L1–3 are as described in Figure 3.

## Data Availability

The data reported in this study are available in the article and Appendix A.

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
