# Peer review of "The ldp1 Mutation Affects the Expression of Auxin-Related Genes and Enhances SAM Size in Rice"

_plants, 2024, doi:10.3390/plants13060759_

Round 1

Reviewer 1 Report

Comments and Suggestions for Authors

Comments to the paper “The ldp1 Mutation Affects the Expression of Auxin-Related Genes and Enhances SAM Size in rice” by Zhanglun Sun et al.

The authors present results of an extensive study addressing panicle type, which represents one of the important factors affecting yield of Oryza sativa L.

In their study the authors used a number of molecular, cytological and genetic approaches such as: comparative RNA-Seq analysis, Weighted Gene Co-expression Network Analysis, scanning electron microscopy and gene mapping.

I fully share the authors conclusion stating that their study provides valuable insights into the molecular regulatory network underlying panicle morphogenesis in Oryza sativa.

The paper is clearly written and requires only a minor revision, which involves corrections of terminology and few linguistic errors. Please see below for details.

1.    The word “turquoise” is sometimes written in lowercase while elsewhere is capitalized. For instance: page 1, line 27, page 1, line 31. The authors should use lowercase form.

2.    I think that the Introduction contains too many details describing morphological development. For instance, page 2, lines 53-57.

3.    I am not sure why while referring to the names of individual genes the authors give their names in uppercase letters in some places while elsewhere the mutants of these genes are given in italicized lowercase letters. For instance: page 2, line 74, page 2, line 75.

4.    Page 7, line 208. Please replace the word “utilized” with “used”. These words are not exactly synonymous.

5.    Page 8, lines 224-225. The words “ion”, “metal”, “cation” should be written in lowercase.

6.    Page 9, line 251. Should be: “represents”.

7.    On page 8, lines 256-260 and elsewhere I found many apostrophes. I think they should be deleted.

8.    Page 10, line 277, Please delete the word “but”.

9.    Page 14, line 408. Should be: “were also”.

10.  In some places, for instance on page 14, line 430 the authors use inconsistent reference format.

11.  Page 14, line 442. Full stop mark is missing after “(stage III)”.

12. Page 16, line 539. Should be: “submitted version”.

Comments on the Quality of English Language

Few minor errors.

Reviewer 2 Report

Comments and Suggestions for Authors

Manuscript titled: The ldp1 Mutation Affects the Expression of Auxin-Related Genes and
Enhances SAM Size in rice
” by  Zhanglun Sun and coworkers, submitted to Plants,  gives an insight into the effect of ldp1 mutation on the expression of key genes in auxin synthesis and signal transduction, which positively influences the size of SAM and thus affects panicle development. 

Rice is one of the most important food crops, providing basic food for about half of the global population. Therefore the improvement in panicle growth, the type of rice inflorescence, especially determination of genes directing the development of panicle architecture can provide insight into the regulatory mechanisms which govern this process and thus enhance breeding.

Understanding of complex process of panicle development has been far from complete and required investigation. Therefore I consider the analysis of the ldp1 mutation which affects panicle development as an important subject of investigations and a good choice of studies from the agriculture  point of view.

The authors demonstrated that the ldp1 mutation leads to substantial changes in WYJ7 gene expression in the earliest stages of floral meristem development, with consequences to panicle morphology. One of the main valuable achievements of this manuscript, demonstrated both by agronomic  measurements and by beautiful images in scanning electron microscopy, was finding that the ldp1 mutation caused an increase in size of the SAM, formation of more primary and secondary branches, and an increase of the number of grains per panicle and grain yield per plant, compared to WYJ7. Another achievement of this manuscript carried out by a detailed genetic analysis was finding  that the ldp1 phenotype is controlled by a recessive monogene located in a 1.2 Mb locus on the seventh chromosome.

Finally the authors, with the help of the RNA-Seq analysis of the three initial stages of panicle development, revealed that key genes in the auxin signaling pathway were differentially expressed during these stages of young inflorescence in WYJ and ldp1.

The authors discussed the regulatory role of ldp1 in the differentiation of panicle primordia and its relation with the endogenous auxin signaling transduction. The authors concluded that although the role of auxin in the regulation of the inflorescence meristem initiation  and growth has been recognized but its dependence on auxin biosynthesis and signaling requires further investigation.

I appreciate very much the clarity of presentation both text and illustrations. I appreciate high quality images from SEM.

Methods are very well performed and described.

Concluding:

I highly evaluate the results obtained in the manuscript titled: “The ldp1 Mutation Affects the Expression of Auxin-Related Genes and Enhances SAM Size in rice” by  Zhanglun Sun and coworkers. In this manuscript,  substantial progress in the elucidation of the function and molecular mechanism of LDP1 during panicle development has been reached.  The authors clarified to a large extend the molecular regulatory network of rice panicle morphogenesis.

I recommend the manuscript to be published in Plants.
